# Functionally specialized human CD4[+] T-cell subsets express physicochemically distinct TCRs

Sofya A Kasatskaya[1,2†], Kristin Ladell[3†], Evgeniy S Egorov[2], Kelly L Miners[3],
Alexey N Davydov[4], Maria Metsger[4], Dmitry B Staroverov[2,5],
Elena K Matveyshina[6], Irina A Shagina[2,5], Ilgar Z Mamedov[2,5], Mark Izraelson[2,5],
Pavel V Shelyakin[1,2], Olga V Britanova[2,5], David A Price[3,7‡],
Dmitriy M Chudakov[1,2,5‡*]

[1]Center of Life Sciences, Skolkovo Institute of Science and Technology, Moscow, Russian Federation; [2]Genomics of Adaptive Immunity Department, Shemyakin-Ovchinnikov Institute of Bioorganic Chemistry, Russian Academy of Sciences, Moscow, Russian Federation; [3]Division of Infection and Immunity, Cardiff University School of Medicine, Cardiff, United Kingdom; [4]Adaptive Immunity Group, Central European Institute of Technology, Brno, Czech Republic; [5]Institute of Translational Medicine, Center for Precision Genome Editing and Genetic Technologies for Biomedicine, Pirogov Russian National Research Medical University, Moscow, Russian Federation; [6]Faculty of Bioengineering and Bioinformatics, Lomonosov Moscow State University, Moscow, Russian Federation; [7]Systems Immunity Research Institute, Cardiff University School of Medicine, Cardiff, United Kingdom

**\*For correspondence:**
chudakovdm@mail.ru

[†]These authors also contributed equally to this work
[‡]These authors also contributed equally to this work

**Competing interests:** The authors declare that no competing interests exist.

**Abstract** The organizational integrity of the adaptive immune system is determined by functionally discrete subsets of CD4[+] T cells, but it has remained unclear to what extent lineage choice is influenced by clonotypically expressed T-cell receptors (TCRs). To address this issue, we used a high-throughput approach to profile the αβ TCR repertoires of human naive and effector/memory CD4[+] T-cell subsets, irrespective of antigen specificity. Highly conserved physicochemical and recombinatorial features were encoded on a subset-specific basis in the effector/memory compartment. Clonal tracking further identified forbidden and permitted transition pathways, mapping effector/memory subsets related by interconversion or ontogeny. Public sequences were largely confined to particular effector/memory subsets, including regulatory T cells (Tregs), which also displayed hardwired repertoire features in the naive compartment. Accordingly, these cumulative repertoire portraits establish a link between clonotype fate decisions in the complex world of CD4[+] T cells and the intrinsic properties of somatically rearranged TCRs.

## Introduction

Adaptive immunity relies on populations of lymphocytes that express somatically rearranged antigen receptors, including CD4[+] T cells, which differentiate from the naive pool into functionally and phenotypically distinct effector/memory subsets that determine how the immune system responds to specific challenges. In the classic dichotomy, mycobacterial and viral infections typically elicit T helper 1 (Th1) cells, which produce interferon (IFN)-γ under the control of T-bet, whereas parasitic infections typically elicit Th2 cells, which produce interleukin (IL)-4, IL-5, and IL-13 under the control of GATA3 and STAT6 (*Mosmann and Coffman, 1989*). Many other subsets have been described in the intervening years (*DuPage and Bluestone, 2016*; *Sallusto, 2016*). The importance of subset

choice as a proximal determinant of response efficacy is apparent from various immune dysregulation syndromes. For example, individuals with Th1 deficiency are predisposed to recurrent bacterial and mycobacterial infections, and individuals with Th17 deficiency are predisposed to chronic muco-candidiasis (*McDonald, 2012*; *Cook and Tangye, 2009*; *Hernández-Santos et al., 2013*). In contrast, systemic autoimmunity is more common in individuals with Th17 overactivity and/or regulatory T-cell (Treg) deficiency (*Osnes et al., 2013*; *Costa et al., 2017*; *Bonelli et al., 2008*; *Miyara et al., 2005*), and allergy is more common in individuals with a similar imbalance between Th2 cells and Tregs (*Bacher and Scheffold, 2018*; *McGee and Agrawal, 2006*; *Finotto, 2008*). Pathogenic and protective roles have also been described for Th9 and Th22 cells in the context of inflammatory skin diseases and various autoimmune conditions, including type I diabetes (*Ryba-Stanisławowska et al., 2016*) and multiple sclerosis (*Rolla et al., 2014*). Similarly, adverse and beneficial outcomes have been associated with the functional attributes of tumor-specific CD4$^+$ T cells, consistently linking Th1-like activity with enhanced survival across a range of cancers (*Protti et al., 2014*). A strictly regulated effector/memory CD4$^+$ T-cell profile is therefore essential for immune function and homeostasis.

Subset choice is dictated by the context of antigen presentation (*Zhu et al., 2010*; *Groom et al., 2012*; *Vroman et al., 2015*; *Baumjohann and Ansel, 2015*; *Waickman et al., 2017*; *Barberis et al., 2018*; *Eisenbarth, 2019*) and potentially by the mode of antigen engagement (*Barberis et al., 2018*; *Adams et al., 2011*; *Wang and Reinherz, 2012*; *Hoffmann et al., 2015*; *Sibener et al., 2018*; *Constant and Bottomly, 1997*; *Corse et al., 2011*). If the latter supposition is correct, then generic molecular signatures may be present among subset-specific repertoires of expressed T-cell receptors (TCRs). To explore this possibility, we systematically deconvoluted the physicochemical and recombinatorial properties of TCRα and TCRβ chains encoded by transcripts isolated from rigorously defined naive and effector/memory subsets of CD4$^+$ T cells. These characteristics provide a broad overview of antigen recognition preferences within a given repertoire and help delineate relatedness among distinct subsets based on patterns of clonotype selection.

Each effector/memory subset was characterized by distinct features that were recapitulated across genetically unrelated donors, indicating a predisposition to certain fate decisions at the level of the somatically rearranged TCR. In line with this notion, similar characteristics were observed in some of the corresponding naive repertoires, most notably those derived from Tregs. Repertoire overlaps further identified effector/memory subsets that were related by common ontogenetic and/or permissible transition pathways. Collectively, these findings map the clonal ancestry and organizational complexity of the human CD4$^+$ T-cell compartment and demonstrate that subset fate is influenced by the structural topography of clonotypically expressed TCRs.

## Results

### Experimental logic and study design

We set out to investigate the naive origins and effector/memory relationships of classically defined CD4$^+$ T-cell subsets in humans. An overview of the experimental workflow designed to capture these complexities is presented in *Figure 1*.

### Effector/memory CD4$^+$ T-cell subsets express physicochemically distinct TCRs

To investigate the TCR repertoires of functionally and phenotypically distinct effector/memory CD4$^+$ T cells, we used polychromatic flow cytometry to identify and sort the commonly recognized Tfh, Th1, Th1-17, Th17, Th22, Th2a, Th2, and Treg subsets from the peripheral blood of healthy donors (n = 5). The gating strategy is described in *Figure 1—figure supplement 1* and *Table 1*. Subset frequencies are listed in *Table 2*. The corresponding TCRα and TCRβ repertoires were obtained from purified mRNA using a high-throughput approach with template switch-based incorporation of unique molecular identifiers (UMIs) as described previously (*Egorov et al., 2015*).

Statistical analyses of the curated TCRα and TCRβ datasets allowed us to describe the somatically rearranged third complementarity-determining region (CDR3) loops in terms of amino acid representation among distinct subsets of effector/memory CD4$^+$ T cells. As in previous studies (*Bolotin et al., 2017*; *Izraelson et al., 2018*; *Egorov et al., 2018*; *De Simone et al., 2019*;

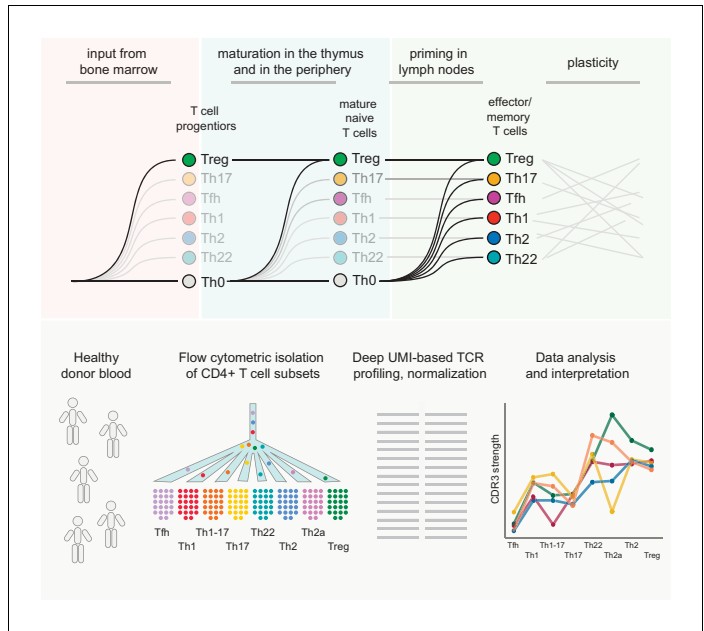

**Figure 1.** Experimental overview. Top: schematic representation of the general questions addressed in this study. Bottom: schematic representation of the experimental pipeline. Naive and effector/memory CD4[+] T-cell subsets were flow-sorted from peripheral blood samples obtained from healthy donors. Repertoire characteristics were extracted from normalized datasets obtained from each subset via high-throughput sequence analysis of all expressed TCRs.

The online version of this article includes the following figure supplement(s) for figure 1:

**Figure supplement 1.** Gating strategy for the identification of effector/memory CD4[+] T-cell subsets.

*Logunova et al., 2020*), we focused on amino acid residues located in the middle of the CDR3 loop, which typically dominate contacts with the peptide component of any cognate pMHC (*Egorov et al., 2018*), and quantified several key physicochemical properties, including hydrophobicity (*Kidera et al., 1985*) and the predicted energy of TCR interactions averaged across diverse pMHCs (*Miyazawa and Jernigan, 1996*; *Kosmrlj et al., 2008*; *Kosmrlj et al., 2010*). This latter parameter provides a generic measure of interaction strength and depends mainly on the prevalence of aromatic and hydrophobic amino acid residues (*Chakrabarti and Bhattacharyya, 2007*).

**Table 1.** Gating strategy for the identification of effector/memory CD4[+] T-cell subsets.

| Gates 1 and 2 | Gate 3 | Gate 4 | Gate 5 | Gate 6 | Gate 7 | Gate 8 | Subset |
|---|---|---|---|---|---|---|---|
| Live single CD3[+] CD14[−] CD19[−] lymphocytes | CD4[+] | Exclude CCR7[+] CD45RA[+] | CD25[high] CD127[low] | | | | Treg |
| | | | CD25[low] CD127[+] | CXCR5[+] | | | Tfh |
| | | | | CCR10[+] | | | Th22 |
| | | | | CXCR5[−] CCR10[−] | CXCR3[+] CCR6[−] | CCR4[−] | Th1 |
| | | | | | CXCR3[−] CCR6[+] | CCR4[+] | Th17 |
| | | | | | CXCR3[+] CCR6[+] | CCR4[−] | Th1-17 |
| | | | | | CXCR3[−] CCR6[−] | CCR4[+] CRTh2[−] | Th2 |
| | | | | | | CCR4[+] CRTh2[+] | Th2a |

See also *Figure 1—figure supplement 1*.

**Table 2.** Frequencies of sorted effector/memory CD4+ T-cell subsets.

| Donor | Tfh | Th1 | Th1-17 | Th17 | Th22 | Th2a | Th2 | Treg |
|---|---|---|---|---|---|---|---|---|
| D1 | 5.44 | 1.91 | 1.44 | 3.06 | 2.60 | 1.04 | 4.86 | 3.99 |
| D2 | 5.82 | 3.29 | 3.50 | 3.14 | 6.64 | 1.53 | 9.24 | 6.92 |
| D3 | 2.05 | 0.19 | 0.31 | 1.31 | 0.81 | 0.26 | 1.93 | 1.84 |
| D4 | 6.70 | 2.33 | 2.11 | 4.22 | 2.02 | 0.57 | 7.19 | 3.95 |
| D5 | 4.39 | 1.16 | 1.17 | 3.32 | 2.12 | 0.82 | 3.96 | 3.99 |
| Mean | 4.88 | 1.78 | 1.71 | 3.01 | 2.84 | 0.84 | 5.44 | 4.14 |
| SD | 1.79 | 1.17 | 1.19 | 1.06 | 2.23 | 0.48 | 2.84 | 1.81 |

Shown as % of live CD3+CD4+CD14−CD19− non-naive cells. Details in **Figure 1—figure supplement 1**.

Hydrophobicity and the propensity to form strong interactions are common but not necessarily determinative features of highly cross-reactive TCRs (*Kosmrlj et al., 2008*; *Kosmrlj et al., 2010*; *Stadinski et al., 2016*).

Although some distinct features, including high scores for hydrophobicity (low Kidera factor 4) and interaction strength in the Treg CDR3β repertoires, were expected from previous studies in mice (*Bolotin et al., 2017*; *Izraelson et al., 2018*; *Logunova et al., 2020*; *Feng et al., 2015*), more unanticipated characteristics were identified among other subsets of effector/memory CD4+ T cells (*Figure 2*). In particular, the Tfh CDR3β repertoires exhibited the lowest averaged scores for hydrophobicity (high Kidera factor 4; *Figure 2C*), interaction strength (*Figure 2D*), and volume (*Figure 2F*, reflects the number of bulky amino acid residues, namely W, R, K, Y, and F [*Shugay et al., 2015*]), and the highest averaged score for surface (*Figure 2E*, provides an in silico predictive measure of amino acid residues that remain unchanged in terms of accessibility and position in the liganded *versus* unliganded state [*Martin and Lavery, 2012*]). These exceptional features suggest that selection into the Tfh subset is driven by highly antigen-specific and minimally cross-reactive TCRs. It is tempting to speculate that such defined molecular patterns, which are mirrored in mature antibody repertoires (*Grimsholm et al., 2020*), act to minimize the risk of autoimmunity, given that Tfh cells play a critical role in the development of B-cell responses.

In addition to Tregs, relatively high numbers of strongly interacting amino acid residues were observed in the Th22, Th2a, and Th2 CDR3β repertoires, which also scored highly in the volume analyses. Of particular note, Th22 cells expressed TCRs with the highest averaged number of random nucleotide (N) additions and the longest averaged CDR3β length, suggesting a distinct but as yet unknown selection process. Consistent physicochemical differences were also apparent between subsets considered as two distinct groups. In general, amino acid characteristics in the Th1/Th1-17/Th17 group resembled those of Tfh cells, whereas amino acid characteristics in the Th22/Th2a/Th2 group resembled those of Tregs (*Figure 2A–F*). Similar patterns were detected in the corresponding CDR3α repertoires (*Figure 2—figure supplement 1*). This overall dichotomy at the population level was clearly visualized using principal component analysis of the cumulative CDR3α and CDR3β repertoires (*Figure 2G*).

Collectively, these data show that subset fate is associated with the physicochemical properties of amino acids in the middle of the CDR3α and CDR3β loops, which typically dominate TCR contacts with the peptide moiety in cognate pMHCs.

## Repertoire diversity varies substantially among effector/memory CD4+ T-cell subsets

In further analyses, we compared repertoire clonality and diversity across the same phenotypically defined subsets of effector/memory CD4+ T cells. Each cloneset was normalized to the lower bound of 16,000 randomly sampled UMI-labeled TCRα or TCRβ cDNA molecules (*Izraelson et al., 2018*). Consistent differences in the corresponding metrics were observed among the various subsets (*Figure 3*).

Prominent clonal expansions, reflected by low normalized Shannon-Wiener indices, were apparent in the Th22 and Th2a subsets, indicating focused antigen-specific proliferation. In contrast, the Tfh subset was highly diverse, incorporating approximately 14,500 distinct sequence variants per

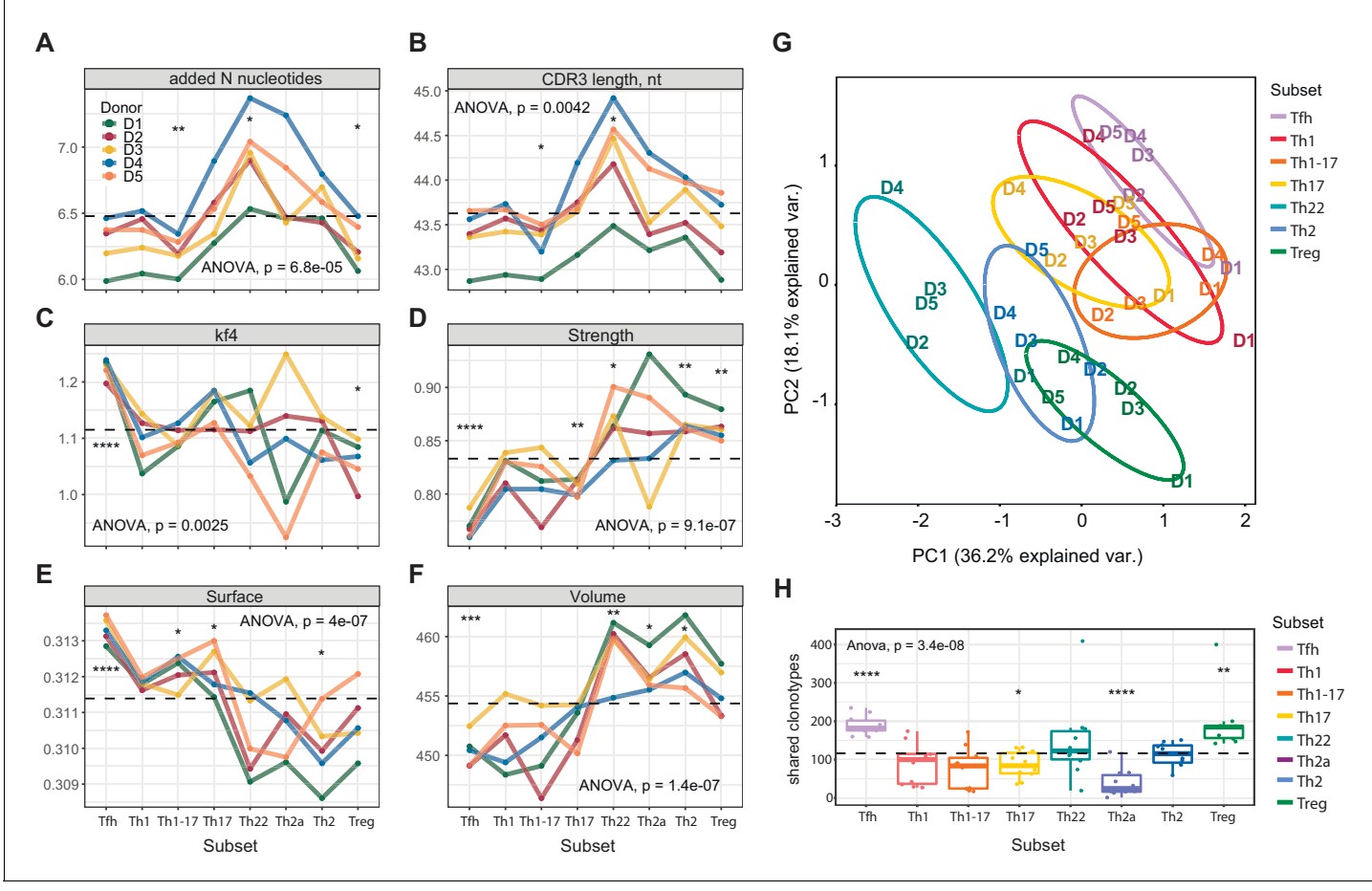

**Figure 2.** Averaged physicochemical characteristics of CDR3β repertoires from effector/memory CD4+ T-cell subsets. (A–F) Averaged physicochemical characteristics were measured for the five amino acids in the middle of the CDR3β sequences obtained from each effector/memory CD4+ T-cell subset (n = 8) from each healthy donor (n = 5). Calculations were weighted by clonotype frequency. Unweighted analyses yielded similar results (data not shown). (A) Non-germline nucleotide (N) additions. (B) CDR3β length (nucleotides). (C) Kidera factor 4 (arbitrary scale). (D) Interaction strength (arbitrary scale). (E) Surface (arbitrary scale). (F) Volume (arbitrary scale). (G) Principal component analysis of the cumulative CDR3α and CDR3β repertoires from each subset of effector/memory CD4+ T cells (n = 28 parameters computed in VDJtools). Top contributing factors to PC1: CDR3β volume, mjenergy, core, beta, length, number of added nucleotides, strength, and alpha. Top contributing factors to PC2: CDR3α disorder, CDR3α Kidera factor 3, CDR3β disorder, CDR3α Kidera factor 1, CDR3α strength, CDR3β Kidera factors 2, 3, 4, and 10, and CDR3β charge. (H) Relative publicity measured for each effector/memory CD4+ T-cell subset as the number of identical or near-identical (maximum n = 1 mismatch) amino acid residue-defined CDR3β variants shared between the top 20,000 most frequent clonotypes in the corresponding repertoires from each pair of donors. Dashed lines indicate means. *p<0.05, **p<0.01, ***p<0.001, and ****p<0.0001 (one-way ANOVA followed by the two-sample Welch t-test with Bonferroni correction for each group *versus* the mean).

The online version of this article includes the following figure supplement(s) for figure 2:

**Figure supplement 1.** Averaged physicochemical characteristics of CDR3α repertoires from effector/memory CD4+ T-cell subsets.

16,000 cDNA molecules. Similar levels of diversity have been observed in umbilical cord blood samples, which almost exclusively contain naive T cells (https://www.biorxiv.org/content/early/2018/09/05/259374). The absence of large clonal expansions among circulating Tfh cells concurs with the findings of a recent study, which also reported greater clonality among donor-matched samples of tonsil-resident Tfh cells (*Brenna et al., 2020*). Relatively high levels of diversity were also observed in the Th1, Th17, and Th2 subsets.

Collectively, these results expose substantial variations in clonality and diversity among effector/memory subsets of CD4+ T cells, likely reflecting distinct selection processes driven by cognate interactions with distinct arrays of pMHCs.

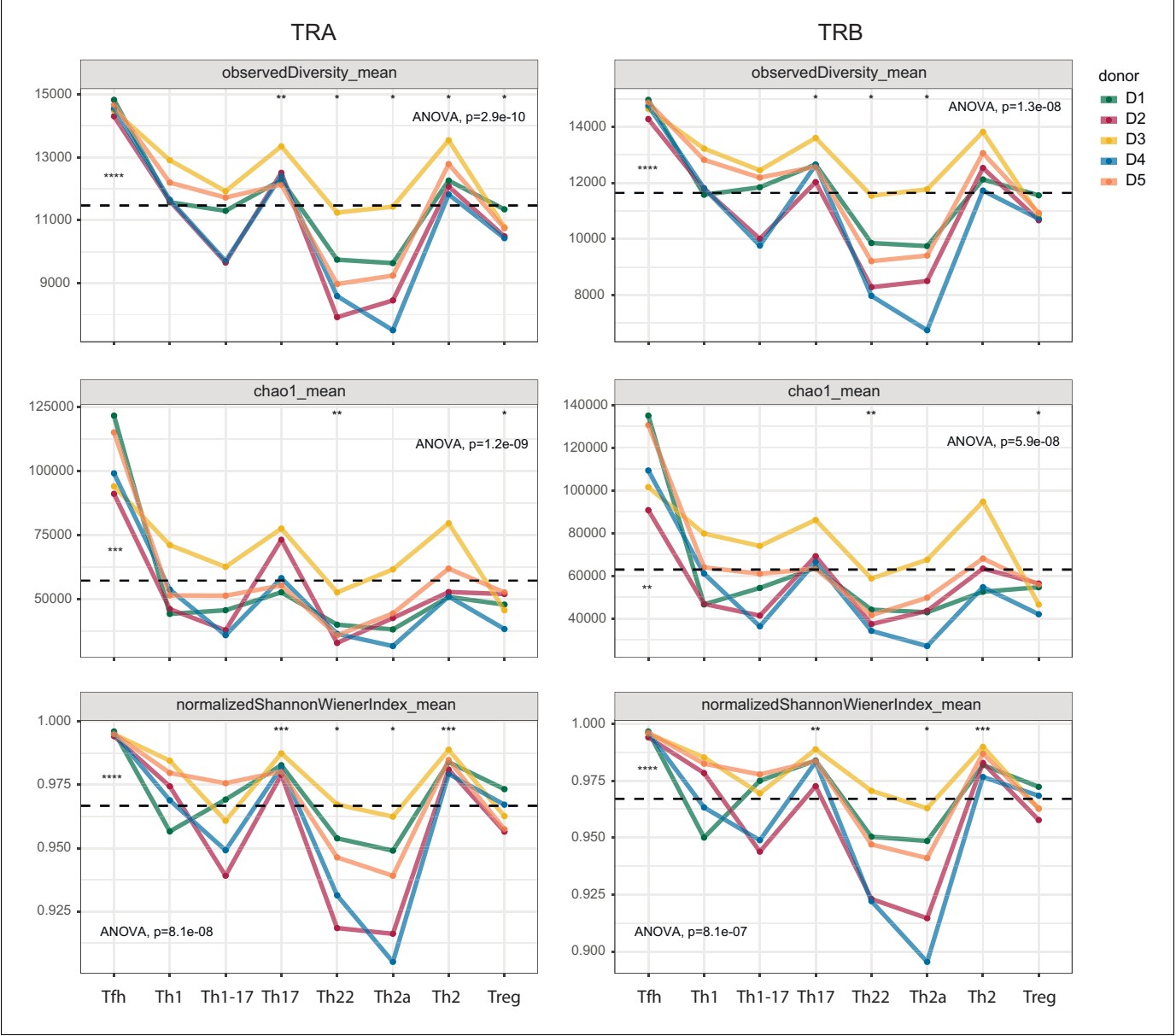

**Figure 3.** Clonality and diversity of effector/memory CD4[+] T-cell subsets. Observed diversity (top), the Chao1 estimator (middle), and the normalized Shannon-Wiener index (bottom) were calculated for each TCRα (left) and TCRβ repertoire (right) obtained from each effector/memory CD4[+] T-cell subset (n = 8) from each healthy donor (n = 5). Dashed lines indicate means. *p<0.05, **p<0.01, ***p<0.001, and ****p<0.0001 (one-way ANOVA followed by the two-sample Welch t-test with Bonferroni correction for each group *versus* the mean).

## Clonal transitions identify related subsets of effector/memory CD4[+] T cells

Effector/memory CD4[+] T cells can switch from one functional subset to another, both in vitro, driven by cytokines, and in vivo, driven by changes in the microenvironment. For example, Th2 cells have been shown to adopt a Th1-like phenotype in mice after infection with lymphocytic choriomeningitis virus, which induces type I and type II IFNs (*Hegazy et al., 2010*). Conversely, Th1 and Th17 cells effectively transitioned into the Th2 subset after transfer into helminth-infected mice, whereas effector Tregs maintained a stable phenotype in the same model (*Panzer et al., 2012*). Previous studies have also shown that human effector Tregs are relatively stable, with rare transitions to the Th1 phenotype occurring only under extreme conditions (*Zhou et al., 2009a*; *Krebs and Steinmetz, 2016*; *McClymont et al., 2011*). However, experiments conducted in vitro or ex vivo are not sufficient to

allow reliable quantitative estimates of plasticity among human effector/memory CD4$^+$ T-cell subsets in vivo.

To address this issue, we measured relative overlap as the number of nucleotide-defined CDR3β clonotypes shared between each pair of subsets in each donor. Similar analyses were conducted using a weighted metric to account for clonotype frequency. The top 20,000 most frequent clonotypes were selected from each TCRβ cloneset to normalize the comparisons (*Figure 4*), and the top 2000 most frequent clonotypes were used to generate the corresponding Cytoscape plots (*Figure 5* and *Figure 5—figure supplements 1–4*). Overall, these analyses revealed prominent clonal exchange among two groups of subsets, namely Th17/Th22/Th2a/Th2 and Th1/Th1-17.

The complementarity and relative functional proximity of the Th17 and Th22 subsets was described previously, albeit without direct evidence of clonal transitions in vivo (*Eyerich et al.,*

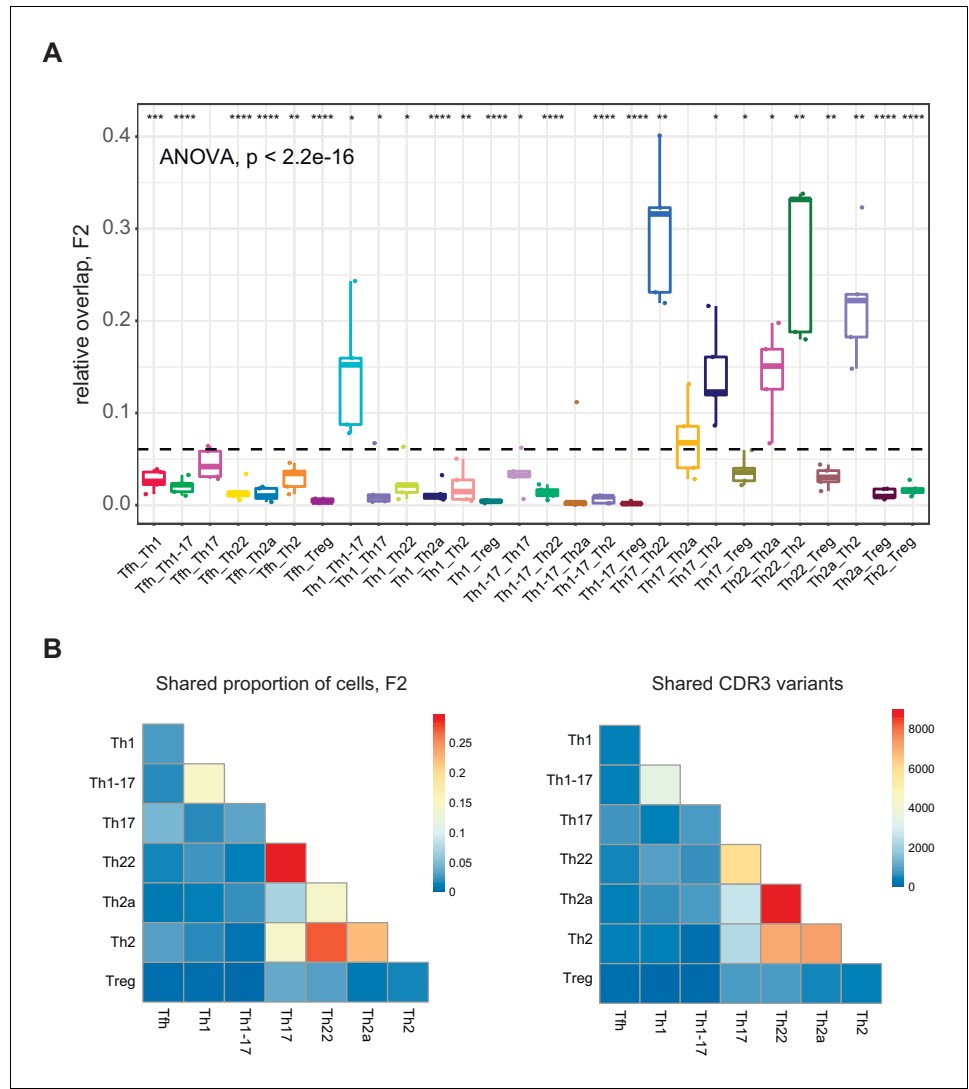

**Figure 4.** Clonotype overlap among effector/memory CD4$^+$ T-cell subsets. (**A**) Relative overlap between nucleotide-defined CDR3β repertoires obtained from donor-matched pairs of effector/memory CD4$^+$ T-cell subsets. Clonotypes were matched on the basis of identical *TRBV* gene segments and identical CDR3β sequences. Data were normalized to the top 20,000 most frequent clonotypes and weighted by clonotype frequency (F2 metric in VDJtools). The dashed line indicates the mean (n = 5 donors). *p<0.05, **p<0.01, ***p<0.001, and ****p<0.0001 (one-way ANOVA followed by the two-sample Welch t-test with Bonferroni correction for each group *versus* the mean). (**B**) Heatmap representations of the weighted overlap (F2 metric in VDJtools, left) and the estimated relative overlap of nucleotide-defined CDR3β clonotypes (calculated via the D metric in VDJtools, right) between donor-matched pairs of effector/memory CD4$^+$ T-cell subsets.

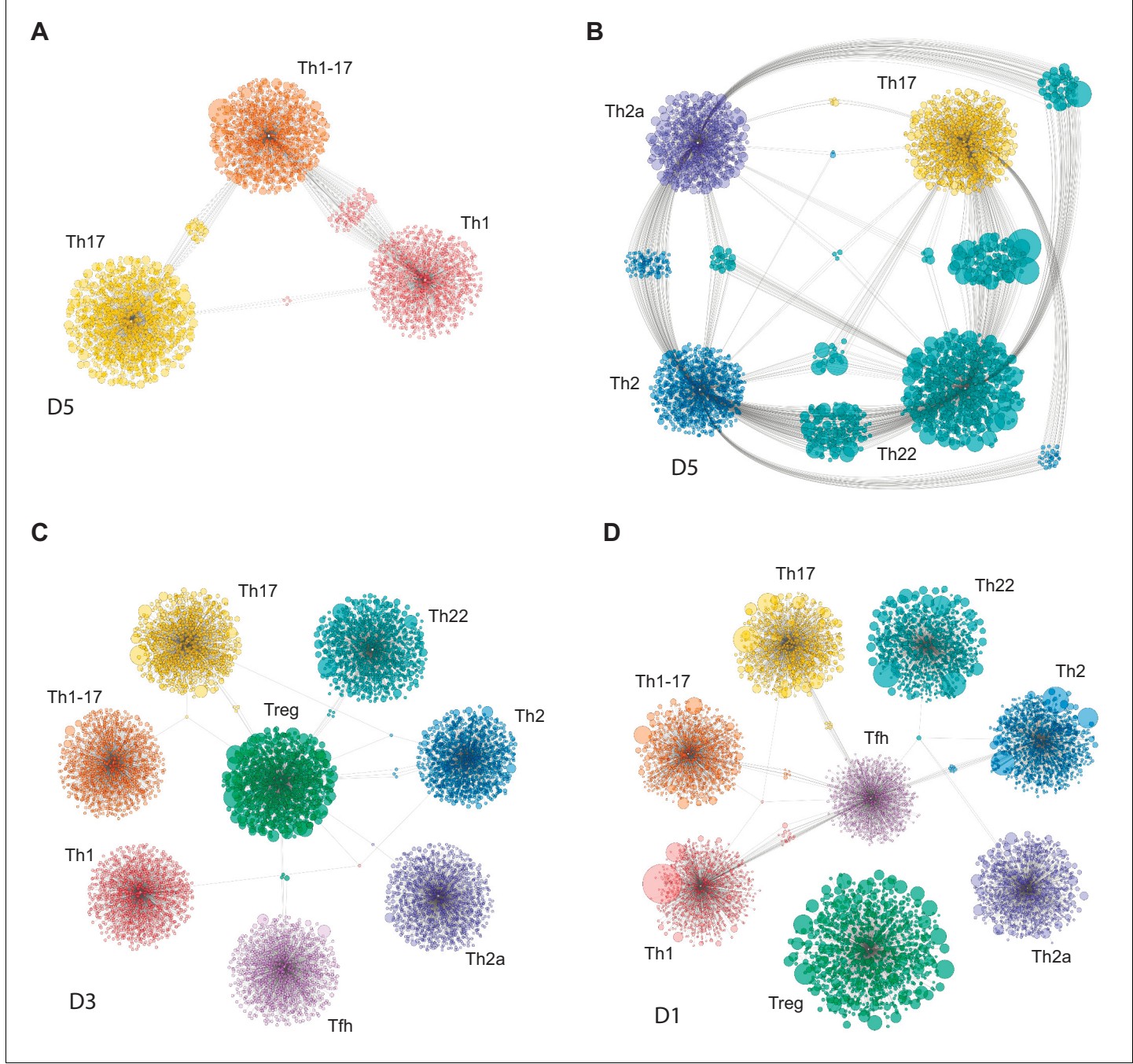

**Figure 5.** Clonal relatedness among effector/memory CD4+ T-cell subsets. Cytoscape network analysis schemes represent the number and size (frequency) of nucleotide-defined clonotype variants shared among the top 2000 most frequent CDR3β clonotypes in each subset. Each bubble represents one CDR3β clonotype. The size of each bubble is proportional to the frequency of each CDR3β clonotype in the corresponding repertoire. Shared clonotypes are depicted as connected clouds among the corresponding subsets. The size of each bubble in these clouds is proportional to the frequency of each CDR3β clonotype averaged across the maternal subsets. Representative plots were selected for illustrative purposes from donors D1, D3, and D5. (A) Th1/Th1-17/Th17. (B) Th17/Th22/Th2a/Th2. (C) Tregs *versus* other subsets. Only clonotypes shared with Tregs are shown. (D) Tfh cells *versus* other subsets. Only clonotypes shared with Tfh cells are shown.

The online version of this article includes the following figure supplement(s) for figure 5:

**Figure supplement 1.** Clonal relatedness among the Th17, Th22, Th2a, and Th2 subsets of effector/memory CD4+ T-cells.

**Figure supplement 2.** Clonal relatedness among the Th1, Th1-17, and Th17 subsets of effector/memory CD4+ T-cells.

**Figure supplement 3.** Clonal relatedness among Tregs and other subsets of effector/memory CD4+ T-cells.

**Figure supplement 4.** Clonal relatedness among Tfh cells other subsets of effector/memory CD4+ T-cells.

*2009*; *Akdis et al., 2012*). However, the close relationships between the Th17 and Th2 subsets and between the Th22 and Th2a/Th2 subsets were unforeseen. Of note, several subsets, including Th17 and Th2 cells, shared large clonal expansions with the Th22 subset (*Figure 5B* and *Figure 5—figure supplement 1*). This observation appears to conflict with the dogma that Th22 cells are stable (*Eyerich et al., 2009*; *Plank et al., 2017*) and suggests that individual clonotypes can seed and/or transition among distinct subsets within the Th17/Th22/Th2a/Th2 group.

It has been suggested previously that Th1-17 cells represent a more mature form of Th17 cells (*Muranski and Restifo, 2013*). In contrast, our findings suggest that Th1-17 cells are more closely related in terms of clonal proximity to Th1 cells rather than Th17 cells. Repertoire overlap between the Th1-17 and Th17 subsets was nonetheless variable among donors, ranging from zero to levels that approximated those observed between the Th1 and Th1-17 subsets (*Figures 4* and *5A*, and *Figure 5—figure supplement 2*).

Collectively, these findings suggest that plasticity is common between certain subsets, such as Th17/Th22 and Th17/Th2, but rare between other subsets, such as Th17/Treg and Th1/Th17 (*Maggi et al., 2012*). In addition, the Tfh and Treg subsets were largely discrete at the clonal level (*Figures 4* and *5*, and *Figure 5—figure supplements 3* and *4*). This latter observation contrasts with previous reports of Treg plasticity (*Zhou et al., 2009a*) but does not exclude the possibility of transient conversions from the committed Treg phenotype (*Yang et al., 2008*; *Voo et al., 2009*).

## Publicity is a notable feature of Tfh cells and Tregs

To extend these analyses, we estimated the extent to which amino acid residue-defined CDR3β clonotypes in each subset were shared among donors, essentially providing a measure of publicity. The top 20,000 most frequent clonotypes were selected from each TCRβ cloneset to normalize the comparisons.

Publicity was observed most commonly among Tfh cells and Tregs, the latter in agreement with previous reports (*Pacholczyk and Kern, 2008*; *Lei et al., 2015*). In contrast, relatively few CDR3β clonotypes in the Th22 and Th2a subsets were shared among donors (*Figure 2H*). These publicity metrics aligned to some extent with subset-specific differences in CDR3β length and the number of N additions (*Figure 2A,B*). One possible explanation for the enrichment of public clonotypes in the Tfh and Treg repertoires lies in the nature of the corresponding antigen-driven selection events. In the case of Tfh cells, common foreign antigens presented in a degenerate manner by MHCs may be recognized predominantly by germline-encoded components of the corresponding TCRs, and in the case of Tregs, common self-derived antigens presented and recognized similarly in the thymus may drive the preferential recruitment of different clonotypes bearing germline-like TCRs.

The relative paucity of N additions in these subsets could reflect low levels of terminal deoxynucleotidyl transferase (TdT) activity, especially among Tregs, some of which arise early in life (*Tulic et al., 2012*; *Coutinho et al., 2005*; *Thiault et al., 2015*; *Darrigues et al., 2018*). A similar phenomenon may likewise explain interindividual differences in publicity, given that all subset-specific effector/memory CD4$^+$ T-cell repertoires in one donor were characterized by low numbers of N additions and relatively short CDR3β loops (*Figure 2A,B*).

## Tregs display similar repertoire features in the naive and effector/memory pools

In general, naive CD4$^+$ T cells are thought to be capable of differentiating into any effector/memory subset from the Th0 state, depending on the composite strength of TCR interactions with cognate pMHCs, costimulatory signals, and the cytokine microenvironment (*Sad and Mosmann, 1994*). However, this paradigm of multipotency has been challenged by the demonstration in several reports that at least some naive CD4$^+$ T cells are predisposed to a specific functional program or even committed to a predetermined fate. This phenomenon was first described for thymic Tregs (tTregs), which maintain a largely stable phenotype in the periphery (*Silva et al., 2016*; *Hoffmann et al., 2006*). At the early immature double-negative stage, thymocytes are already predisposed to the Treg lineage via epigenetic modifications and increased expression of FoxP3 (*Ohkura et al., 2012*; *Arvey et al., 2015*). Other inputs are then required to confirm this commitment, including signals delivered by the IL-2 receptor and intermittent stimulation via high-affinity TCRs (*Levine et al., 2014*). A similar process of agonist-driven selection has been described for thymic Th17 cells in mice

(*Marks et al., 2009*). Accordingly, subset fate may be imprinted at the progenitor stage (*Feng et al., 2015*), during thymic development (*Li and Rudensky, 2016*), after thymic emigration and before determinative antigen encounter (*Fink, 2013*), and/or during the key priming event that signals expansion and maturation (*Figure 1*).

On the basis of these considerations, we reasoned that certain subset-specific repertoire features, at least in the case of Tregs, could be conserved between the corresponding naive and effector/memory pools. To investigate this prediction, we profiled the TCRα and TCRβ repertoires of naive CD4[+] T cells flow-sorted as recent thymic emigrants (RTEs) (*Kilpatrick et al., 2008*), mature naive T cells, or naive Tregs from the peripheral blood of healthy donors (total, n = 12; twin pairs, n = 5).

The naive Treg CDR3β repertoires were enriched for bulky, hydrophobic, and strongly interacting amino acid residues compared with the corresponding RTE and mature naive T-cell repertoires (*Figure 6A* and *Figure 6—figure supplement 1*). These observations are consistent with potent agonist-driven selection in the thymus (*Feng et al., 2015*; *Jordan et al., 2001*). In addition, naive Tregs expressed TCRs with shorter CDR3α and CDR3β loops. Similar features were observed in the effector/memory Treg compartment (*Figure 2*).

To confirm and extend these findings, we conducted similar analyses of naive CD4[+] T-cell subsets flow-sorted as Th1-like cells (non-Treg CCR4[−]CXCR3[+]), Th2-like cells (non-Treg CCR4[+]CXCR3[−]), and Tregs (CD25[high]CD127[low]) from healthy donors (n = 4) matching those shown in *Figure 2*. The corresponding non-Treg CCR4[−]CXCR3[−] and non-Treg CCR4[+]CXCR3[+] populations were analyzed in parallel for comparative purposes.

The naive Treg CDR3β repertoires were again enriched for bulky, hydrophobic, and strongly interacting amino acid residues compared with the other naive subset-specific CDR3β repertoires

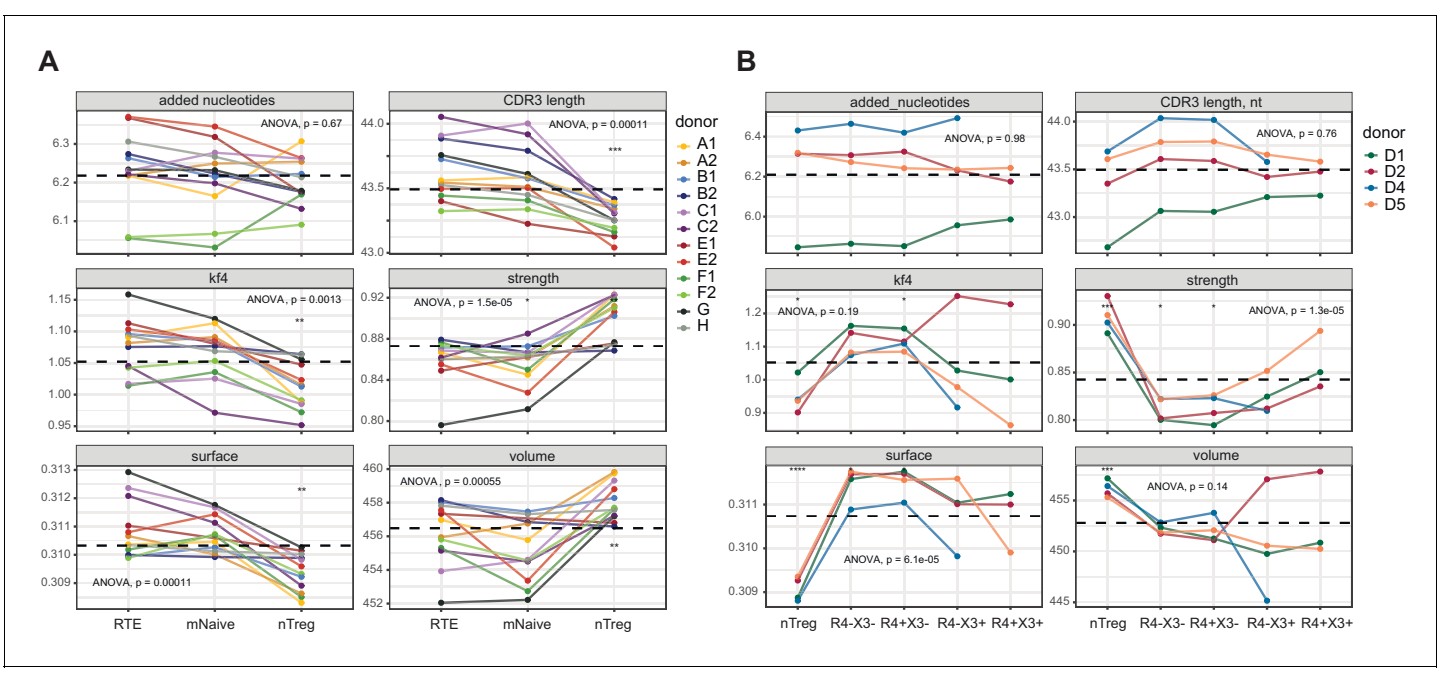

**Figure 6.** Averaged physicochemical characteristics of CDR3β repertoires from naive CD4[+] T-cell subsets. (A) Repertoire analysis of RTEs (CD25[−]CD31[+]), mature naive T cells (mNaive; CD25[−]CD31[−]), and naive Tregs (nTreg; CD25[high]) from healthy donors (n = 12). Matched letters in the key indicate twin pairs. (B) Repertoire analysis of naive Th1-like cells (non-Treg CCR4[−]CXCR3[+]), naive Th2-like cells (non-Treg CCR4[+]CXCR3[−]), naive Tregs (CD25[high]CD127[low]), and the corresponding non-Treg CCR4[−]CXCR3[−] and non-Treg CCR4[+]CXCR3[+] populations from healthy donors (n = 4) matching those shown in *Figure 2*. Averaged physicochemical characteristics were measured for the five amino acids in the middle of the CDR3β sequences obtained from each naive CD4[+] T-cell subset. Calculations were weighted by clonotype frequency. Parameter details as in *Figure 2*. Dashed lines indicate means. *p<0.05, **p<0.01, ***p<0.001, and ****p<0.0001 (one-way ANOVA followed by the two-sample Welch t-test with Bonferroni correction for each group *versus* the mean).

The online version of this article includes the following figure supplement(s) for figure 6:

**Figure supplement 1.** Averaged physicochemical characteristics of CDR3α repertoires from naive CD4[+] T-cell subsets.

**Figure supplement 2.** Gating strategy for the identification of naive CD4[+] T-cell subsets.

(*Figure 6B*). In addition, both naive CXCR3$^+$ subsets were characterized by increased numbers of strongly interacting amino acid residues compared with the corresponding naive CXCR3$^-$ subsets. A similar dichotomy has been reported for naive CD8$^+$ T cells (*De Simone et al., 2019*). However, the naive Th1-like and naive Th2-like CDR3β repertoires were generally physicochemically distinct from the corresponding effector/memory CDR3β repertoires, in contrast to Tregs (*Figure 2*).

Collectively, these results indicate that distinct repertoire features are hardwired in the Treg lineage during thymic selection, whereas other subset-specific repertoires are generally shaped later in ontogeny, most likely driven by naive CD4$^+$ T-cell interactions with cognate pMHCs.

## Discussion

In this study, we used polychromatic flow cytometry and an unbiased high-throughput sequencing approach to probe the ontogeny and relatedness of commonly recognized effector/memory CD4$^+$ T-cell subsets via in-depth analyses of clonotypically expressed TCRs. We found that each subset-specific repertoire was characterized by distinct physicochemical and recombinatorial features that were highly reproducible across multiple donors. Importantly, these differences were multivariate, such that each subset displayed an array of repertoire characteristics, which in aggregate delineated the spectrum of preferred TCRs.

The CDR3α and CDR3β repertoires of effector/memory Tregs contained the highest numbers of hydrophobic and strongly interacting amino acid residues. These features were recapitulated in the corresponding naive Treg repertoires, suggesting that lineage fate was predetermined by selection events in the thymus (*Feng et al., 2015*; *Jordan et al., 2001*). Similar physicochemical characteristics have been associated previously with highly cross-reactive TCRs (*Kosmrlj et al., 2008*; *Kosmrlj et al., 2010*; *Stadinski et al., 2016*). However, naive and effector/memory Tregs also expressed TCRs with relatively short CDR3α and CDR3β loops, which might limit steric flexibility and thereby enhance the specificity of antigen-driven selection (*Li and Rudensky, 2016*; *Bacher et al., 2016*; *Su et al., 2016*; *Spence et al., 2018*; *Akkaya et al., 2019*). Such composite properties are compatible with an inherent predilection for self-derived peptides tempered by a capacity for ligand discrimination. Of note, the effector/memory Treg subset as identified in this study potentially incorporated both thymus-derived and peripherally induced Tregs (*Hoffmann et al., 2006*). In line with the possibility of mixed origins, higher numbers of N additions were detected in the naive Treg repertoires compared with the effector/memory Treg repertoires, potentially indicating the long-term persistence of early fetal Tregs (*Booth et al., 2010*).

Substantial heterogeneity is thought to exist in the Treg lineage (*Sawant and Vignali, 2014*). For example, peripheral interconversion between Th17 cells and Tregs has been observed in the presence of IL-6 and TGF-β1 (*Murphy and Stockinger, 2010*), and a loss of Foxp3 expression along with regulatory functions has been observed in the context of lymphopenia (*Tang et al., 2008*; *Lathrop et al., 2008*). In a more recent evaluation of this latter phenomenon, however, the ex-Foxp3$^+$ cells that accumulated under lymphopenic conditions were not *bona fide* Tregs, but rather descendants of non-Tregs that transiently expressed Foxp3 (*Miyao et al., 2012*). Partial transition from the Treg subset has also been associated with the Tr1 phenotype, distinguished by high production levels of IL-10 (*Häringer et al., 2009*). In contrast, we found little evidence of plasticity among effector/memory Tregs, suggesting a largely fixed lineage choice, irrespective of potentially diverse origins.

Our analysis of circulating Tfh (cTfh) cells likely included migratory components of the Th1-like, Th17-like, Th2-like, and follicular regulatory (Tfr) subpopulations of Tfh cells (*Bentebibel et al., 2013*; *Morita et al., 2011*; *Linterman et al., 2011*; *Chung et al., 2011*; *Maceiras et al., 2017*; *Yang et al., 2019*). Unexpectedly in light of this potential heterogeneity, we found that the cTfh CDR3α and CDR3β repertoires were characterized by extreme features, including the lowest numbers of bulky, hydrophobic, and strongly interacting amino acid residues, with low dispersion among donors and little overlap with other subsets. These characteristics were further associated with short CDR3α and CDR3β loops. Accordingly, cTfh cells formed a distinct cluster in the principal component analysis, closest to the Th1 subset. This configuration suggests a high degree of antigen specificity with minimal cross-reactivity (*Kosmrlj et al., 2008*; *Kosmrlj et al., 2010*; *Stadinski et al., 2016*). It is tempting to speculate that such features are required to prevent the induction of autoantibody responses. In support of this hypothesis, remarkably similar features are acquired

progressively in the B-cell repertoire during the course of affinity maturation, reflecting intense negative selection of cross-reactive antibody variants (*Grimsholm et al., 2020*). Our data also suggest that non-hydrophobic contacts underpin antigen specificity in the context of high-affinity interactions between Tfh cell-expressed TCRs and cognate pMHCs (*Fazilleau et al., 2009a*).

In the periphery, cTfh cells survey multiple tissue sites and respond swiftly to previously encountered antigens, providing a systemic mirror of germinal center reactions after exiting the inceptive lymph node (*Shulman et al., 2013*; *Vella et al., 2019*). We found no evidence of clonal expansions in the cTfh repertoires of healthy donors, likely reflecting the random nature of recirculation and the consequent sampling of mixed specificities. This interpretation concurs with the findings of a recent study, in which clonality was low among cTfh cells and high among tonsillar Tfh cells (*Brenna et al., 2020*). Network analysis further revealed that cTfh-expressed TCRs were largely subset-specific and rarely exhibited clonal transitions. This observation again concurs with previous work (*Brenna et al., 2020*). Accordingly, cTfh cells appear to represent a distinct lineage rather than a differentiation step in the progressive maturation of other subsets, as proposed in some earlier models (*Fazilleau et al., 2009b*; *Vinuesa et al., 2016*).

Th22 cells are typically found in the skin, where they play a key role in wound healing (*Alabbas et al., 2018*) and epidermal immunity (*Eyerich et al., 2009*). Pathogenic activity has also been ascribed to this subset in the contexts of multiple sclerosis (*Rolla et al., 2014*), rheumatoid arthritis (*Miyazaki et al., 2018*), and chronic skin graft-*versus*-host disease (*Gartlan et al., 2018*). On the basis of in vitro studies, Th22 cells are thought to exhibit plasticity with Th1 and possibly with Th2 cells (*Plank et al., 2017*). Our systematic analysis of plasticity in vivo does not support this view. Instead, we found that Th22 cells shared large expansions of unique clonotypes with the Th17, Th2a, and Th2 subsets. This pattern was recapitulated across all donors. Cluster feature analysis nonetheless suggested the existence of clonotypically discrete populations of *bona fide* Th22 cells.

Th1 and Th2 cells are widely considered to be the most stably differentiated subsets of effector/memory CD4$^+$ T cells (*Zhou et al., 2009b*), both in vitro and in vivo (*Murphy and Stockinger, 2010*; *Murphy et al., 1996*; *Messi et al., 2003*; *Brown et al., 2015*). However, some central memory Th1 cells can produce large quantities of IL-4 under Th2-polarizing conditions (*Rivino et al., 2004*). Conversely, murine Th2 cells primed in vivo can acquire the ability to produce IFN-γ as well as IL-4 (*Hegazy et al., 2010*), whereas human Th2 cells seem to be more immutable (*Messi et al., 2003*). In functional terms, Th2 cells are clearly defined by the production of IL-4, but in phenotypic terms, the key lineage-defining markers remain a matter of debate, with most laboratories using either CCR4$^+$CCR6$^-$ or CCR6$^-$CRTh2$^+$ as the critical parameters. To bypass this controversy, we analyzed CCR4$^+$CCR6$^-$CRTh2$^-$ (Th2) cells and CCR4$^+$CCR6$^-$CRTh2$^+$ (Th2a) cells separately. The core repertoire of the Th2a subset was unique, implying a specialized function, but interestingly, both the Th2a and Th2 subsets shared clonal expansions with the Th22 subset. This unexpected finding nonetheless aligns with current revisions of the classic paradigm toward a more plastic view of Th1 cells (*Leipe et al., 2020*).

In contrast to Th1 and Th2 cells, Th17 cells and Tregs, including naturally occurring and peripherally induced Tregs, are thought to be inherently plastic (*Geginat et al., 2014*), especially in mice (*Cohen et al., 2011*). For example, murine and human Th17 cells differentiated in vivo can be induced to adopt a Th1-like or Th1-17-like phenotype in vitro (*Lee et al., 2009*; *Annunziato et al., 2007*; *Hirota et al., 2011*), and human Th17 cells migrating to sites of inflammation can acquire a Th1-like phenotype, characterized by the expression of CD161 as well as CCR6 (*Maggi et al., 2012*). We found that Th17 cells most commonly shared clonal expansions with Th22 cells, which also shared clonal expansions with Th2a and Th2 cells. Little is known about such transitions, in part because TGF-β1 promotes the development of Th17 cells and inhibits the development of Th2 cells, which are consequently separated in most differentiation schemes (*Muranski and Restifo, 2013*). Further studies are therefore required to interpret these findings in mechanistic terms. Of note, we did not analyze Th9 cells, which appear to derive from Th17 cells under inflammatory conditions (*Beriou et al., 2010*) and are thought to be relatively unstable (*Schlapbach et al., 2014*).

The development of Th17 cells and Tregs in the thymus is linked due to shared microenvironmental factors that favor commitment to both lineages. These cells may also derive from common thymic progenitors (*Yang et al., 2008*). In vivo, Th17 cells have been shown to acquire certain regulatory features, including the ability to produce IL-10 under the influence of IL-12 or IL-27

(*Heinemann et al., 2014*). However, we found no evidence of interconversion between Th17 cells and Tregs, at least within the effector/memory pool of CD4$^+$ T cells.

Th1-17 cells are thought to represent a more mature form of Th17 cells (*Muranski and Restifo, 2013*). Unexpectedly, we found that Th1-17 cells shared few or no clonotypes with Th17 cells, whereas clonal overlap was common between Th1 and Th1-17 cells. In line with this dichotomy, Th17 cells, but not Th1-17 cells, shared large clonal expansions with Th22 cells. The intermediate nature of Th1-17 cells has been predicted using computational models (*Puniya et al., 2018*) and observed directly in vitro (*Zielinski et al., 2012*). Ex vivo, Th1-17 cells are characterized by the coproduction IL-17, IL-22, and IFN-γ (*Duhen and Campbell, 2014*). Our data suggest that Th1-17 cells are more closely related to Th1 cells rather than Th17 cells, but nonetheless, the core Th1-17 repertoires were largely unique, suggesting that a majority of these cells occupy a distinct lineage and do not simply represent a maturation stage in the development of Th1 or Th17 cells.

Effector/memory CD4$^+$ T-cell subsets are classified according to distinct patterns of cytokine production, reflecting differential expression of various master transcription factors. However, these profiles were largely established on the basis of in vitro studies, and consequently, our current understanding of subset phylogeny is most likely an oversimplification (*Zhu et al., 2010*). Mixed and unexplored subsets therefore almost certainly exist in vivo, reflecting nuances in the epigenetic landscape (*Allan et al., 2012*) and the relative activities of master regulators (*Kanhere et al., 2012*; *Aune et al., 2009*). Greater understanding of these complexities could inform efforts to develop more effective therapies for autoimmune diseases (*Ryba-Stanisławowska et al., 2016*; *Rolla et al., 2014*; *Walker and von Herrath, 2016*) and cancer (*Kreiter et al., 2015*; *Borst et al., 2018*; *Wei et al., 2017*), as well as better targeted vaccines (*Misiak et al., 2017*). In this context, our data provide an important step on the path to systematic deconvolution of the CD4$^+$ T-cell compartment, specifically via the demonstration that subset fate is associated with the non-random selection of clonotypes expressing physicochemically distinct TCRs.

## Materials and methods

### Samples

Venous blood samples were collected from healthy adult donors (n = 17) directly into heparinized syringes or Vacutainer EDTA Tubes (BD Biosciences). Peripheral blood mononuclear cells (PBMCs) were isolated via density gradient centrifugation over Ficoll-Paque (PanEco) or Histopaque-1077 (Sigma-Aldrich). Ethical approval was granted by the institutional review committees at Cardiff University School of Medicine (16/55) and the Pirogov Russian National Research Medical University (2017/52). All donors provided written informed consent in accordance with the Declaration of Helsinki.

### Flow cytometric sorting of effector/memory CD4$^+$ T-cell subsets

PBMCs were stained immediately after isolation (n = 5 donors) with LIVE/DEAD Fixable Aqua (Thermo Fisher Scientific) and the following directly conjugated monoclonal antibodies: anti-CCR6–PE (clone 11A9), anti-CCR7–PE-Cy7 (clone 3D12), anti-CD14–V500 (clone M5E2), anti-CD19–V500 (clone HIB19), and anti-CRTh2–PE-CF594 (clone BM16) from BD Biosciences; anti-CCR4–BV605 (clone L291H4), anti-CD3–APC-Fire750 (clone SK7), anti-CD25–BV711 (clone MA251), anti-CD45RA–PE-Cy5 (clone HI100), anti-CD127–BV421 (clone A019D5), and anti-CXCR5–BV785 (clone J252D4) from BioLegend; anti-CCR10–APC (clone 314305) and anti-CXCR3–FITC (clone 49801.111) from R&D Systems; and anti-CD4–PE-Cy5.5 (clone S3.5) from Thermo Fisher Scientific. The gating strategy is described in *Figure 1—figure supplement 1* and *Table 1*. Subsets were flow-sorted at >98% purity after exclusion of naive CCR7$^+$CD45RA$^+$ events from the Aqua$^-$CD3$^+$CD4$^+$CD14$^-$CD19$^-$ gate as Tfh cells (CXCR5$^+$), Th1 cells (non-Tfh/Th22/Treg CCR4$^-$CCR6$^-$CXCR3$^+$), Th1-17 cells (non-Tfh/Th22/Treg CCR4$^-$CCR6$^+$CXCR3$^+$), Th17 cells (non-Tfh/Th22/Treg CCR4$^+$CCR6$^+$CXCR3$^-$), Th22 cells (CCR10$^+$), Th2a cells (non-Tfh/Th22/Treg CCR4$^+$CCR6$^-$CRTh2$^+$CXCR3$^-$), Th2 cells (non-Tfh/Th22/Treg CCR4$^+$CCR6$^-$CRTh2$^-$CXCR3$^-$), or Tregs (CD25$^{high}$CD127$^{low}$) using a modified FACSAria II (BD Biosciences). All cells (n = 6,000–150,000 per subset) were sorted directly into RLT buffer (Qiagen) containing 1% 2-mercaptoethanol (Sigma-Aldrich). Subset frequencies are listed in *Table 2*. Acquisition and post-sort data were analyzed using FlowJo software version 10.6.1 (Tree Star).

## Flow cytometric sorting of naive CD4$^+$ T-cell subsets

To identify RTEs, mature naive T cells, and naive Tregs in the CD4$^+$ lineage, PBMCs were stained immediately after isolation (n = 12 donors) with the following directly conjugated monoclonal antibodies: anti-CD4–PE (clone 13B8.2) and anti-CD27–PE-Cy5 (clone O323) from Beckman Coulter; and anti-CD25–eFluor450 (clone BC96), anti-CD31–PE-Cy7 (clone WM59), and anti-CD45RA–FITC (clone JS-83) from eBioscience. The gating strategy was described previously (*Egorov et al., 2018*). Subsets were flow-sorted at >98% purity from the CD4$^+$CD27$^+$CD45RA$^+$ gate as RTEs (CD25$^-$CD31$^+$), mature naive T cells (CD25$^-$CD31$^-$), or naive Tregs (CD25$^{high}$) using a FACS Aria III (BD Biosciences). To identify naive Th1-like cells, naive Th2-like cells, and naive Tregs in the CD4$^+$ lineage, PBMCs were stained immediately after isolation (n = 4 donors) with LIVE/DEAD Fixable Aqua (Thermo Fisher Scientific) and the following directly conjugated monoclonal antibodies: anti-CCR7–PE-Cy7 (clone 3D12), anti-CD8–V500 (clone RPA-T8), anti-CD14–V500 (clone M5E2), and anti-CD19–V500 (clone HIB19) from BD Biosciences; anti-CCR4–BV605 (clone L291H4), anti-CD3–APC-Fire750 (clone SK7), anti-CD25–BV711 (clone MA251), anti-CD45RA–PE-Cy5 (clone HI100), anti-CD95–PE (clone DX2), and anti-CD127–BV421 (clone A019D5) from BioLegend; anti-CXCR3–FITC (clone 49801.111) from R&D Systems; and anti-CD4–PE-Cy5.5 (clone S3.5) from Thermo Fisher Scientific. The gating strategy is described in *Figure 6—figure supplement 2*. Subsets were flow-sorted at >98% purity from the Aqua$^-$CD3$^+$CD4$^+$CD8$^-$CD14$^-$CD19$^-$CCR7$^+$CD45RA$^+$CD95$^-$ gate as naive Th1-like cells (non-Treg CCR4$^-$CXCR3$^+$), naive Th2-like cells (non-Treg CCR4$^+$CXCR3$^-$), or naive Tregs (CD25$^{high}$CD127$^{low}$), alongside the corresponding non-Treg CCR4$^-$CXCR3$^-$ and non-Treg CCR4$^+$CXCR3$^+$ populations, using a modified FACS Aria II (BD Biosciences). All cells (n = 260-150,000 per subset) were sorted directly into RLT buffer (Qiagen) containing 1% 2-mercaptoethanol (Sigma-Aldrich). Subset frequencies are listed in *Table 3*. Acquisition and post-sort data were analyzed using FlowJo software version 10.6.1 (Tree Star).

## TCR sequencing and data analysis

TCRα and TCRβ cDNA libraries were prepared using a Human TCR Kit (MiLaboratory LLC) with template switch-based incorporation of UMIs as described previously (*Egorov et al., 2015*). Libraries were sequenced in paired-end mode (150 + 150 bp) on a NextSeq500 (Illumina). Raw sequence data were analyzed using MIGEC software version 1.2.9 (*Shugay et al., 2014*). Briefly, UMI sequences were extracted from demultiplexed data using the Checkout utility, yielding sample barcode matches in ~90% of cases. Data were then assembled using the erroneous UMI filtering option in the Assemble utility. For most tasks, the minimum required number of reads per UMI was set at 1. For analyses of overlap and publicity, which are sensitive to even minor cross-sample contaminations, the minimum required number of reads per UMI was set at 3 (*Egorov et al., 2015*). In-frame TCRα and TCRβ repertoires were extracted using MiXCR software version 2.1.1 (*Bolotin et al., 2017*; *Bolotin et al., 2018*; *Bolotin et al., 2015*). At a threshold of 3 reads per UMI, the number of obtained UMI-labeled cDNA molecules per repertoire per sample ranged from 5300 to 303,500, and the number of CDR3 clonotype variants at the nucleotide level per repertoire per sample ranged from 1200 to 83,200. Normalization, data transformation, in-depth analyses, and statistical calculations were performed using R scripts and VDJtools software version 1.2.1 (*Shugay et al., 2015*). Analyses of averaged CDR3 characteristics were weighted by the abundance of each clonotype in each sample. Basic characteristics included CDR3 length, the number of N additions, interaction strength, hydrophobicity (Kidera factor 4), volume, and surface, which were selected in previous

**Table 3.** Frequencies of sorted naive CD4$^+$ T-cell subsets.

| Donor | Th1-like CCR4$^-$CXCR3$^+$ | Th2-like CCR4$^+$CXCR3$^-$ | CCR4$^-$ CXCR3$^-$ | CCR4$^+$ CXCR3$^+$ | Treg CD25$^{high}$ CD127$^{low}$ |
|---|---|---|---|---|---|
| D1 | 1.75 | 5.46 | 44.80 | 0.23 | 0.73 |
| D2 | 0.77 | 6.77 | 20.40 | 0.32 | 0.57 |
| D3 | 0.15 | 5.67 | 42.60 | 0.19 | 1.70 |
| D4 | 0.16 | 6.33 | 33.10 | 0.05 | 1.11 |

Shown as % of live CD3$^+$CD4$^+$CD8$^-$CD14$^-$CD19$^-$ naive cells. Details in **Figure 6—figure supplement 2**.

analyses of various somatically rearranged lymphocyte receptor datasets (*Izraelson et al., 2018*; *Egorov et al., 2018*; *Davydov et al., 2018*). The amino acid properties used in these analyses can be viewed at https://github.com/mikessh/vdjtools/blob/master/src/main/resources/profile/aa_property_table.txt. The strength feature reflects the predicted sum of interaction affinities between pairs of amino acids at the TCR-pMHC interface, Kidera factor 4 reflects the abundance of hydrophobic amino acids on an inverted scale, and the surface characteristic reflects the relative abundance of amino acids with no predicted changes in accessibility during TCR engagement with cognate pMHCs. Amino acid hierarchies by probability of active involvement at the protein-protein interface or conformational stability relative to the native form in the absence of an interaction were derived from previous work (*Martin and Lavery, 2012*), in which extensive cross-docking experiments were performed across 198 proteins and 300 partners in silico to infer the general roles of amino acids at protein-protein interfaces. Physicochemical characteristics were calculated and averaged for the five amino acid residues located in the middle of each CDR3 loop, which are most likely to contact the peptide epitope in any cognate pMHC (*Egorov et al., 2018*). Principal component analysis was performed using 28 parameters computed as the average across each CDR3α and CDR3β cloneset: Kidera factors (n = 10), strength, mjenergy, count (CDR3 length), NDN length, number of N insertions, vdins, djins, core, rim, volume, polarity, disorder, surface, alpha, beta, turn, charge, and hydropathy (VDJtools software version 1.2.1). No significant variations in V/J segment use were detected among subsets (data not shown). Network visualization was performed using Cytoscape (https://cytoscape.org). Repertoire overlap was analyzed using the unweighted D (reflecting the proportion of shared clonotypes between paired repertoires) and weighted F2 (reflecting the proportion of shared T cells between paired repertoires) metrics in VDJtools software version 1.2.1.

## Quantification and statistical analysis

Statistical analyses were performed on processed datasets in R. Multiple parameter inferences were estimated using ANOVA if the data were distributed normally or the Kruskal-Wallis test if any of the data were not distributed normally. The corresponding p values were calculated using the two-sample Welch t-test or the Wilcoxon rank sum test. The false discovery rate was controlled using Benjamini-Hochberg correction unless stated otherwise. Post-hoc tests were performed using the ggpubr package (https://CRAN.R-project.org/package=ggpubr).

## Acknowledgements

This work was supported by grants from the Ministry of Science and Higher Education of the Russian Federation (075-15-2019-1789) and the Wellcome Trust (100326/Z/12/Z).

## Additional information

### Funding

| Funder | Grant reference number | Author |
| --- | --- | --- |
| Ministry of Science and Higher Education | 075-15-2019-1789 | Dmitriy M Chudakov |
| Wellcome Trust | 100326/Z/12/Z | David A Price |

The funders had no role in study design, data collection and interpretation, or the decision to submit the work for publication.

### Author contributions

Sofya A Kasatskaya, Data curation, Visualization, Methodology, Writing - original draft, Writing - review and editing; Kristin Ladell, Data curation, Investigation, Methodology, Writing - original draft; Evgeniy S Egorov, Kelly L Miners, Maria Metsger, Dmitry B Staroverov, Irina A Shagina, Ilgar Z Mamedov, Investigation, Methodology; Alexey N Davydov, Data curation, Investigation, Methodology; Elena K Matveyshina, Mark Izraelson, Pavel V Shelyakin, Data curation; Olga V Britanova, Data curation, Supervision, Writing - original draft; David A Price, Conceptualization, Supervision, Funding acquisition, Writing - original draft, Writing - review and editing; Dmitriy M Chudakov,

Conceptualization, Supervision, Funding acquisition, Validation, Investigation, Visualization, Writing - original draft, Project administration, Writing - review and editing

## Author ORCIDs
Elena K Matveyshina (ID) http://orcid.org/0000-0003-4641-4906
David A Price (ID) https://orcid.org/0000-0001-9416-2737
Dmitriy M Chudakov (ID) https://orcid.org/0000-0003-0430-790X

## Ethics
Human subjects: ethical approval was granted by the institutional review committees at Cardiff University School of Medicine (reference number 16/55) and the Pirogov Russian National Research Medical University (protocol number 2017/52), and all donors provided written informed consent in accordance with the Declaration of Helsinki.

## Decision letter and Author response
Decision letter https://doi.org/10.7554/eLife.57063.sa1
Author response https://doi.org/10.7554/eLife.57063.sa2

## Additional files
### Supplementary files
• Transparent reporting form

### Data availability
All extracted repertoires and metadata are available in Figshare: https://figshare.com/s/2145b1b16c6854445af7 and https://figshare.com/s/84ec5f412356afb0536d. Deep TCR profiling data were deposited in the GEO under accession code GSE158848.

The following datasets were generated:

| Author(s) | Year | Dataset title | Dataset URL | Database and Identifier |
|---|---|---|---|---|
| Kasatskaya SA, Ladell K, Miners KL, Davydov AN, Britanova OV, Price DA, Chudakov DM | 2020 | Human effector/memory CD4+ T cell subsets: deep TCR profiling. | https://www.ncbi.nlm.nih.gov/geo/query/acc.cgi?acc=GSE158848 | NCBI Gene Expression Omnibus, GSE158848 |
| Kasatskaya SA, Ladell K, Miners KL, Davydov AN, Britanova OV, Price DA, Chudakov DM | 2020 | TCR repertoires in human CD4+ T cell subsets: effector/memory subsets, TCR alpha & TCR beta chain sequencing | https://figshare.com/s/2145b1b16c6854445af7 | figshare, 10.6084/m9.figshare.2145b1b16c6854445af7 |
| Kasatskaya SA, Ladell K, Egorov ES, Miners KL, Staroverov DB, Britanova OV, Price DA, Chudakov DM | 2020 | TCR beta repertoires in naive CD4+ T cell subsets in identical twin donor samples | https://figshare.com/s/84ec5f412356afb0536d | figshare, 10.6084/m9.figshare.84ec5f412356afb0536d |

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
