## [Decision Letter]

**Acceptance summary:**

This study uses T cell receptor sequencing to probe the structure of the functional T cell pool in healthy human blood. The key findings are (1) distinctive physicochemical, recombinational, and clonality characteristics for many of the repertoires, and (2) conserved and stereotyped patterns of clonal sharing between the subsets within donors and of public amino acid chains between donors. The results shed significant new light on how T cell dependent immunity is organized.

**Decision letter after peer review:**

Thank you for submitting your article "Origin and plasticity of human CD4^+^ T cell subsets tracked via TCRs" for consideration by *eLife*. Your article has been reviewed by three peer reviewers, and the evaluation has been overseen by a Reviewing Editor and Satyajit Rath as the Senior Editor. The following individuals involved in review of your submission have agreed to reveal their identity: Benjamin Chain (Reviewer #2); Rob J de Boer (Reviewer #5).

The reviewers have discussed the reviews with one another and the Reviewing Editor has drafted this decision to help you prepare a revised submission.

This study uses T cell receptor sequencing to probe the structure of the functional T cell pool in healthy human blood. In particular, the manuscript reports a characterization of TCR α and β repertoire features for eight effector/memory CD4^+^ T cell subsets defined by flow cytometry across 5 donors, and of several naive CD4 subsets across 12 donors. The key findings are (1) distinctive physicochemical, recombinational, and clonality characteristics for many of the repertoires, and (2) conserved and stereotyped patterns of clonal sharing between the subsets within the donors and of public amino acid chains between donors. All reviewers agree that the results shed significant new light on how T cell dependent immunity is organized. However, there are still some points that we would like to see addressed in the manuscript.

1) It would be beneficial to the readers if the authors could rewrite parts of the manuscript to make the story more coherent. Currently, the manuscript reads as very descriptive, and there is little by the way of hypothesis (except perhaps that Tregs are antigen-selected in the thymus more than are Teff). As a result, the study reads a bit like a series of disconnected observations, and it is hard to build up a clear unified message. Perhaps the authors could also add a table or cartoon of the novel classification suggested by the paper, in which the TCR properties and the repertoire overlaps are matched as much as possible. Currently, the first and the second part of the paper are disconnected, and if both parts are true, these results should be related and suggest a similar classification.

2) There is a lack of clarity about the statistical analysis of the differences between the populations which weakens the impact of the conclusions. For example, in Figure 1, it is difficult to get an indication of the extent of the variation that exists, and the biological amplitude of the effect. It is not clear if the parametric ANOVA is the right test here. It would be interesting to do a non-parametric test, based on shuffling of the repertoire labels, for example, and see the extent of variation observed by chance. It would be interesting also to see the results using unique sequences only (not weighted for frequency), perhaps a supplemental data. The magnitude of the effects is quite small – in the order of half a nucleotide length, for example. It would be useful to get a much better feel for the real variation in the population. Similar points apply to other figures.

3) The strikingly lower diversity of TH22 and Th2a in Figure 2 seems interesting. Could the authors provide a bit more detail on what is driving these changes? A few very large clones? A different clonal distribution? Or, fewer singlets?

4) With regard to the sequence characteristics that differ between subsets: it would be good to confirm that these are not due to differences in V/J gene usage, since the J gene in particular can contribute substantially to the CDR3 and the location of the 5 residue “central” window will overlap to varying degrees with this germline-derived sequence depending on the CDR3 length. It would also be good to rule out the possibility that there are recurrent, semi-invariant amino acid motifs present in the sequences, as opposed to generic sequence biases arising from physicochemical differences.

5) With regard to differential clonal dynamics: How can the authors rule out the possibility that the apparent differences in clonality arise from differences in mRNA expression of the TCR chains, leading to varying numbers of cDNA templates per cell?

6) The sharing data (e.g. Figure 3) is a central point of the paper, and is everywhere interpreted as evidence for plasticity, which is not necessarily true. Alternatively, the results could also mean that populations which share more sequences are derived form a common progenitor – in other words a lineage tree effect. It is not clear how the authors distinguish between these two possibilities, and a more detailed discussion would be helpful with this regards.

7) With regard to sharing, could the author discuss possibility for convergent recombination to lead to identical nucleotide sequences and hence apparent clone sharing, particularly for sequences that are close to germline. Also, are V/J genes included in the definition of "nucleotide clonotypes" or just the CDR3 sequence?

8) Figure 4 is very pretty but unfortunately, very opaque. The legend does not really explain at all what is being shown, or what it purports to show.

9) A general question: Are the TCR differences between the phenotypes large enough to classify a cell based upon its TCR into a phenotype? Probably not, which would mean that the results are (very interesting) trends.

[Editors' note: further revisions were suggested prior to acceptance, as described below.]

Thank you for resubmitting your work entitled "Origin and plasticity of human CD4^+^ T cell subsets tracked via TCRs" for further consideration by *eLife*. Your revised article has been evaluated by Satyajit Rath (Senior Editor) and a Reviewing Editor.

The manuscript has been improved but there are some remaining issues that need to be addressed before acceptance, as outlined below:

First, the evidence for plasticity is not convincing enough, and it is only one possible hypothesis, for which the manuscript does not provide a decisive and a strong support. Please change the title of the manuscript to reflect this.

Reviewer #2:

I raised a number a number of concerns in my first review, most of which I feel have not been addressed. In particular, I remain unconvinced that this paper has anything to do with plasticity. I still struggle to understand Figure 5. or what it adds to the narrative.

The title remains misleading – I don't think the paper says anything about either the origin or the plasticity of the cells.

However, I agree with the reviewer 5 comment "The fact that different CD4 T cell phenotypes tend to use different TCRs is a very innovative result ", and of general interest. This is an important result, even though the effect size is small, and worthy of publication. But it is not at all clear from the title that this is what the paper is about.

Reviewer #3:

My concerns have been addressed.

Reviewer #5:

Reading the revised manuscript convinced me again that this a truly interesting paper with several surprising results.

My only major recommendation would be to add a sentence saying that V and J usage are not different between the subsets.

---

## [Author Response]

This study uses T cell receptor sequencing to probe the structure of the functional T cell pool in healthy human blood. In particular, the manuscript reports a characterization of TCR α and β repertoire features for eight effector/memory CD4^+^ T cell subsets defined by flow cytometry across 5 donors, and of several naive CD4 subsets across 12 donors. The key findings are (1) distinctive physicochemical, recombinational, and clonality characteristics for many of the repertoires, and (2) conserved and stereotyped patterns of clonal sharing between the subsets within the donors and of public amino acid chains between donors. All reviewers agree that the results shed significant new light on how T cell dependent immunity is organized. However, there are still some points that we would like to see addressed in the manuscript.1) It would be beneficial to the readers if the authors could rewrite parts of the manuscript to make the story more coherent. Currently, the manuscript reads as very descriptive, and there is little by the way of hypothesis (except perhaps that Tregs are antigen-selected in the thymus more than are Teff). As a result, the study reads a bit like a series of disconnected observations, and it is hard to build up a clear unified message. Perhaps the authors could also add a table or cartoon of the novel classification suggested by the paper, in which the TCR properties and the repertoire overlaps are matched as much as possible. Currently, the first and the second part of the paper are disconnected, and if both parts are true, these results should be related and suggest a similar classification.

Thank you for the deep comment.

We worked through the whole manuscript to add our considerations where appropriate, and to some extent to better link the effector and naive parts of the manuscript.

We have also included a new section on experimental logic and workflow in the Results, which includes a graphical summary (new Figure 1).

However, it looks like we cannot currently build exact correlations between naive and effector subsets beyond Tregs and, to some extent, CXCR3-positive subsets. We also do not think that we could provide better classification and visualization compared with the current Figures 2, 4, and 5.

2) There is a lack of clarity about the statistical analysis of the differences between the populations which weakens the impact of the conclusions. For example, in Figure 1, it is difficult to get an indication of the extent of the variation that exists, and the biological amplitude of the effect. It is not clear if the parametric ANOVA is the right test here. It would be interesting to do a non-parametric test, based on shuffling of the repertoire labels, for example, and see the extent of variation observed by chance. It would be interesting also to see the results using unique sequences only (not weighted for frequency), perhaps a supplemental data. The magnitude of the effects is quite small – in the order of half a nucleotide length, for example. It would be useful to get a much better feel for the real variation in the population. Similar points apply to other figures.

Effects are relatively small, that’s correct, and so it is important that we observe the very same differences in unrelated healthy donors. To describe the level of deviation from the normal distribution, we built QQ plots for chosen physicochemical CDR3α/β characteristics, Author response image 1. Here we used the comparison to all samples as a post-hoc test instead of multiple pairwise comparisons among the subset groups. This analysis shows that the distribution is normal for most parameters, and thus ANOVA should be the right test.

Non-parametric post-hoc test had little difference from the pairwise assessment of groups with a parametric post-hoc test.

**Author response table 1. resptable1:** 

property	column used	compare to	subset	p adjusted (BH method)	p	signif	test	p adjusted (BH method)	p	signif	test
cdr3_length	zscore	.all.	Th2	5.1e-01	0.43258	ns	T-test	0.600	0.4481	ns	Wilcoxon
cdr3_length	zscore	.all.	Tfh	2.4e-01	0.14003	ns	T-test	0.370	0.2333	ns	Wilcoxon
cdr3_length	zscore	.all.	Th2a	7.4e-01	0.72099	ns	T-test	0.770	0.7726	ns	Wilcoxon
cdr3_length	zscore	.all.	Th1	4.3e-01	0.34963	ns	T-test	0.770	0.7451	ns	Wilcoxon
cdr3_length	zscore	.all.	Th1-17	9.2e-02	0.03070	*	T-test	0.250	0.0829	ns	Wilcoxon
cdr3_length	zscore	.all.	TREG	4.3e-01	0.34053	ns	T-test	0.600	0.4481	ns	Wilcoxon
cdr3_length	zscore	.all.	Th17	7.5e-01	0.75446	ns	T-test	0.670	0.5633	ns	Wilcoxon
cdr3_length	zscore	.all.	Th22	1.1e-01	0.04145	*	T-test	0.120	0.0171	*	Wilcoxon
added_nucleotides	zscore	.all.	Th2	3.1e-01	0.21194	ns	T-test	0.330	0.1935	ns	Wilcoxon
added_nucleotides	zscore	.all.	Tfh	1.4e-01	0.07180	ns	T-test	0.330	0.1699	ns	Wilcoxon
added_nucleotides	zscore	.all.	Th2a	3.7e-01	0.26285	ns	T-test	0.330	0.1935	ns	Wilcoxon
added_nucleotides	zscore	.all.	Th1	2.6e-01	0.16583	ns	T-test	0.540	0.3860	ns	Wilcoxon
added_nucleotides	zscore	.all.	Th1-17	2.2e-02	0.00365	**	T-test	0.120	0.0251	*	Wilcoxon
added_nucleotides	zscore	.all.	TREG	1.1e-01	0.04517	*	T-test	0.310	0.1485	ns	Wilcoxon
added_nucleotides	zscore	.all.	Th17	7.4e-01	0.70804	ns	T-test	0.670	0.5633	ns	Wilcoxon
added_nucleotides	zscore	.all.	Th22	7.6e-02	0.01900	*	T-test	0.073	0.0060	**	Wilcoxon
strength	zscore	.all.	Th2	1.1e-02	0.00135	**	T-test	0.160	0.0362	*	Wilcoxon
strength	zscore	.all.	Tfh	1.8e-06	7.3e-08	****	T-test	0.049	0.0021	**	Wilcoxon
strength	zscore	.all.	Th2a	4.3e-01	0.34108	ns	T-test	0.470	0.3118	ns	Wilcoxon
strength	zscore	.all.	Th1	3.9e-01	0.28682	ns	T-test	0.610	0.4700	ns	Wilcoxon
strength	zscore	.all.	Th1-17	2.6e-01	0.16136	ns	T-test	0.370	0.2333	ns	Wilcoxon
strength	zscore	.all.	TREG	1.5e-02	0.00219	**	T-test	0.310	0.1386	ns	Wilcoxon
strength	zscore	.all.	Th17	1.0e-02	0.00104	**	T-test	0.310	0.1292	ns	Wilcoxon
strength	zscore	.all.	Th22	1.0e-01	0.03680	*	T-test	0.190	0.0556	ns	Wilcoxon
surface	zscore	.all.	Th2	9.2e-02	0.03016	*	T-test	0.120	0.0251	*	Wilcoxon
surface	zscore	.all.	Tfh	2.0e-06	1.3e-07	****	T-test	0.049	0.0019	**	Wilcoxon
surface	zscore	.all.	Th2a	2.4e-01	0.14789	ns	T-test	0.330	0.1935	ns	Wilcoxon
surface	zscore	.all.	Th1	1.2e-01	0.05902	ns	T-test	0.670	0.5392	ns	Wilcoxon
surface	zscore	.all.	Th1-17	5.5e-02	0.01270	*	T-test	0.310	0.1485	ns	Wilcoxon
surface	zscore	.all.	TREG	3.1e-01	0.21607	ns	T-test	0.430	0.2785	ns	Wilcoxon
surface	zscore	.all.	Th17	1.1e-01	0.04722	*	T-test	0.330	0.1814	ns	Wilcoxon
surface	zscore	.all.	Th22	1.6e-01	0.08850	ns	T-test	0.190	0.0603	ns	Wilcoxon
volume	zscore	.all.	Th2	5.5e-02	0.01168	*	T-test	0.120	0.0208	*	Wilcoxon
volume	zscore	.all.	Tfh	4.6e-03	0.00039	***	T-test	0.120	0.0229	*	Wilcoxon
volume	zscore	.all.	Th2a	7.8e-02	0.02114	*	T-test	0.310	0.1485	ns	Wilcoxon
volume	zscore	.all.	Th1	1.4e-01	0.06982	ns	T-test	0.310	0.1120	ns	Wilcoxon
volume	zscore	.all.	Th1-17	1.2e-01	0.05435	ns	T-test	0.190	0.0603	ns	Wilcoxon
volume	zscore	.all.	TREG	5.3e-01	0.46185	ns	T-test	0.670	0.6131	ns	Wilcoxon
volume	zscore	.all.	Th17	2.3e-01	0.13201	ns	T-test	0.480	0.3294	ns	Wilcoxon
volume	zscore	.all.	Th22	3.8e-02	0.00721	**	T-test	0.120	0.0140	*	Wilcoxon
kf4	zscore	.all.	Th2	6.3e-01	0.56570	ns	T-test	0.770	0.7726	ns	Wilcoxon
kf4	zscore	.all.	Tfh	3.5e-07	7.3e-09	****	T-test	0.049	0.0031	**	Wilcoxon
kf4	zscore	.all.	Th2a	6.3e-01	0.57609	ns	T-test	0.680	0.6387	ns	Wilcoxon
kf4	zscore	.all.	Th1	5.0e-01	0.41381	ns	T-test	0.670	0.6131	ns	Wilcoxon
kf4	zscore	.all.	Th1-17	4.3e-01	0.32935	ns	T-test	0.670	0.5879	ns	Wilcoxon
kf4	zscore	.all.	TREG	9.2e-02	0.02848	*	T-test	0.170	0.0431	*	Wilcoxon
kf4	zscore	.all.	Th17	1.2e-01	0.05305	ns	T-test	0.310	0.1386	ns	Wilcoxon
kf4	zscore	.all.	Th22	7.1e-01	0.66248	ns	T-test	0.670	0.5879	ns	Wilcoxon

3) The strikingly lower diversity of TH22 and Th2a in Figure 2 seems interesting. Could the authors provide a bit more detail on what is driving these changes? A few very large clones? A different clonal distribution? Or, fewer singlets?

As indicated in the text: “Prominent clonal expansions, reflected by low normalized Shannon-Wiener indices, were apparent in the Th22 and Th2a subsets, indicating focused antigen-specific proliferation.”

4) With regard to the sequence characteristics that differ between subsets: it would be good to confirm that these are not due to differences in V/J gene usage, since the J gene in particular can contribute substantially to the CDR3 and the location of the 5 residue “central” window will overlap to varying degrees with this germline-derived sequence depending on the CDR3 length. It would also be good to rule out the possibility that there are recurrent, semi-invariant amino acid motifs present in the sequences, as opposed to generic sequence biases arising from physicochemical differences.

Several approaches could be utilized to estimate the contribution of recurrent or semi-invariant sequences/motifs. For one, the prevalence of semi-invariant sequences could be reflected in the analysis of CDR3 length and N insertions, as shorter sequences appear in repertoires with higher frequencies. The distribution of N inserted nucleotides is shown in Figure 2A. Another approach could be to search for functional invariant sequences. We performed a search for classical TRA CDR3 described previously for human iNKT and MAIT cells:

iNKT:

CVVIDRGSTLGRLYF, CVVSDRGSTLGRLYF

MAIT:

CAVKDSNYQLIF, CAGMDSNYQLIF, CASIDSNYQLIF, CAAMDSNYQLIF, CAAEDSNYQLIF, CAVVDSNYQLIF, CAVRDSNYQLIF, CAVMDSSYKLIF, CAVMDSSYKLIF, CAVMDSSYKLIF, CAVRDGDYKLSF, CAVSDSNYQLIF, CAVMDSNYQLIF, CAFMDSNYQLIF

Cumulative frequencies of such invariant TRA CDR3s in repertoires were 0.17% for MAIT and 0.14% for the iNKT cells. There were no significant differences in frequencies among subsets. Therefore, this analysis did not reveal any substantial bias in invariant TCRs distribution between functional subsets.

Also, we have initially spent substantial time trying to find any dependencies in V and J usage between the subsets, and have not found any. In general, V and J usages are distributed randomly across the subsets, so this should not be a prominent contributing factor:

**Author response image 2. respfig2:** 

5) With regard to differential clonal dynamics: How can the authors rule out the possibility that the apparent differences in clonality arise from differences in mRNA expression of the TCR chains, leading to varying numbers of cDNA templates per cell?

Thank you for the comment. We cannot formally completely exclude this possibility, but it seems unlikely, given that each subset was sorted rigorously on defined phenotypic parameters.

Hypothetically, distinct TCR mRNA expression levels between clones could to some extent influence the apparent differences in observed clonality, which remains beyond the scopes of the current work. However, in reality, all our experience with UMI-based TCR profiling shows that such influence should be negligible.

We would prefer not to overload the manuscript with these considerations, but ready to add the appropriate comment to the manuscript if Editor considers it appropriate.

6) The sharing data (e.g. Figure 3) is a central point of the paper, and is everywhere interpreted as evidence for plasticity, which is not necessarily true. Alternatively, the results could also mean that populations which share more sequences are derived form a common progenitor – in other words a lineage tree effect. It is not clear how the authors distinguish between these two possibilities, and a more detailed discussion would be helpful with this regards.

Formally, we cannot distinguish between these alternative scenarios in our analysis. However, we believe that sharing between the top-2000 most frequent clonotypes is much more likely explained by the current plasticity. Long-term evolution from a common progenitor would most probably result in prominent clonal expansions observed in a particular subset, so that we would not sample the same clone among dominating in two distinct subsets.

As well as with previous question, we are ready to add the appropriate comment to the manuscript if Editor considers it appropriate.

7) With regard to sharing, could the author discuss possibility for convergent recombination to lead to identical nucleotide sequences and hence apparent clone sharing, particularly for sequences that are close to germline. Also, are V/J genes included in the definition of "nucleotide clonotypes" or just the CDR3 sequence?

V-genes were included in the analysis. We have added a note to the legend of Figure 4. We cannot completely exclude the possibility of convergent recombination here. However, its input should be negligible, especially in the top-2000 clonotypes. Please note that increased plasticity is observed predominantly between the subsets with long CDR3s and high count of added N nucleotides (Th22-Th2-Th2a-Th17). i.e. this is not about convergence.

8) Figure 4 is very pretty but unfortunately, very opaque. The legend does not really explain at all what is being shown, or what it purports to show.

Thank you. We do have the high resolution figures and will definitely take care of the quality during the final stages of manuscript preparation together with the Editors. We have also worked on the figure legend to make it more informative.

9) A general question: Are the TCR differences between the phenotypes large enough to classify a cell based upon its TCR into a phenotype? Probably not, which would mean that the results are (very interesting) trends.

No, of course not, not only because the differences are small, but also because it is a clone-specific story. This only works as average characteristics of the subset-specific repertoires, likely reflecting other factors that determine fate decisions, including the context of antigen presentation, as highlighted in the Introduction.

As we indicate now in the text (first chapter of Results):

“It should be noted that the above characteristics were observed for the averaged, cumulative portrait of many TCR variants representing each subset. Each particular T cell with low number of N-added nucleotides and strongly interacting aminoacids in the middle of short CDR3 may belong to any Th subset, but more probably to Tfh, Th1, or Th1-17.”